# Astaxanthin Alleviates Foam Cell Formation and Promotes Cholesterol Efflux in Ox-LDL-Induced RAW264.7 Cells via CircTPP2/miR-3073b-5p/ABCA1 Pathway

**DOI:** 10.3390/molecules28041701

**Published:** 2023-02-10

**Authors:** Zhexiao Zhang, Yunmei Qiu, Wanzhi Li, Anyang Tang, Hang Huang, Wanyi Yao, Huawen Li, Tangbin Zou

**Affiliations:** 1The First Dongguan Affiliated Hospital, Guangdong Medical University, Dongguan 523710, China; 2Dongguan Key Laboratory of Environmental Medicine, School of Public Health, Guangdong Medical University, Dongguan 523808, China; 3Huangpu District Center for Disease Control and Prevention, Guangzhou 510700, China

**Keywords:** astaxanthin, cholesterol efflux, macrophage, circTPP2, ABCA1

## Abstract

Atherosclerosis (AS) is a common cardiovascular disease and remains the leading cause of death in the world. It is generally believed that the deposition of foam cells in the arterial wall is the main cause of AS. Moreover, promoting cholesterol efflux and enhancing the ability of reverse cholesterol transport (RCT) can effectively inhibit the formation of foam cells, thereby preventing the occurrence and development of AS. Astaxanthin, with a powerful antioxidant ability, has a potential role in the prevention of atherosclerosis, but how it works in preventing atherosclerosis remains unknown. Here, our experimental results suggest that astaxanthin can upregulate the expression of circular RNA tripeptidyl-peptidase II (circTPP2) and eventually promote cholesterol efflux by modulating ATP-binding cassette subfamily A member 1 (ABCA1). The expression of ABCA1 was significantly suppressed after knocking down circTPP2 in macrophage-derived foam cells. In addition, the experimental results showed that circTPP2 could downregulate the expression of microRNA-3073b-5p (miR-3073b-5p), and ABCA1 was identified as the target gene of miR-3073b-5p. In conclusion, the circTPP2/miR-3073b-5p/ABCA1 axis may be the specific mechanism of astaxanthin promoting cholesterol efflux.

## 1. Introduction

Cardiovascular diseases are still the leading cause of death globally, currently killing more people each year than any other cause of death [1,2]. Atherosclerosis (AS), with an increasing morbidity and mortality in developing countries, is also the primary disease in China, threatening the health of the general population [3]. The deposition of foam cells in the arterial wall is the underlying pathophysiological basis of AS, which leads to a series of changes such as damage in the endothelial cells and smooth muscle of the artery wall and an excessive inflammatory fibroproliferative response [4]. The formation of macrophage-derived foam cells is due to the excessive intake of oxidized low-density lipoprotein (ox-LDL) and the accumulation of cholesterol esters. Low-density lipoprotein (LDL) transports the endogenous cholesterol and lipids from the liver to the periphery, while high-density lipoprotein (HDL), with a variety of anti-atherosclerotic activities, is responsible for reverse cholesterol transport (RCT), transporting the cholesterol and lipids from the periphery to the liver for catabolism [5,6]. RCT is a process of transporting excess peripheral cholesterol from the surrounding tissue back to the liver and secreting it into bile and feces through HDL, which can effectively inhibit the occurrence and development of AS [7,8]. Cholesterol efflux is the first and key step of RCT, and promoting the occurrence of this process is beneficial to inhibit the formation and deposition of macrophage-derived foam cells. ATP-binding cassette subfamily A member 1 (ABCA1), ATP-binding cassette subfamily G member 1 (ABCG1), and scavenger receptor class B type I (SR-BI) play an indispensable role in promoting cholesterol efflux from macrophages [9,10,11]. With the increase in their expression, they can promote cholesterol efflux, reduce the accumulation of cholesterol in macrophages, and finally alleviate the symptoms of AS [12,13].

Astaxanthin (AST), a non-vitamin A pro-carotenoid widely used in salmonid and crustacean aquaculture, is known to have powerful antioxidant and anti-inflammatory properties [14,15]. AST also effectively preserves the structural integrity of the mitochondrial membrane, maintains a high mitochondrial membrane potential, and stimulates cell respiration, thus improving its energy production efficiency [16]. As a nutritional dietary supplement, the appropriate daily intake of AST is 4–40 mg/day [17]. AST was reported to achieve the highest absorbance rate when consuming fatty acid-containing food simultaneously [18]. Excessive oxidative stress and the inflammatory cascade play a critical role in the occurrence and development of AS. AST attracted extensive attention during the past decades as a dietary protective factor against AS [19]. Kumar et al. [20] showed that rat long-term AST intake resulted in a dramatic serum reduction in total cholesterol (TC) and low-density lipoprotein cholesterol (LDL-C) levels and an elevation of high-density lipoprotein cholesterol (HDL-C) concentration. Meanwhile, AST upregulated the levels of HDL-C and adiponectin, as well as inhibited the levels of ox-LDL and apolipoprotein (apoA-I) [14,21,22]. Furthermore, Iizuka et al. [23] pointed out that ABCA1/G1′s upregulation represented a key mechanism of apoA-I/HDL-mediated cholesterol efflux and reversion of AS progression. However, the underlying mechanisms of AST-mediated promoting cholesterol efflux remain largely unknown.

Circular RNA (circRNA), a special type of noncoding RNA (ncRNA) generated by reverse splicing of pre-mRNA, usually regulates protein interactions and participates in ribosomal RNA processing and translation processes [24,25,26,27]. Recent studies have proposed that circRNAs may be involved in the development of RCT by acting as miRNA sponges [28,29,30]. Previous studies from our group suggested that AST can effectively reduce THP-1 cell damage induced by ox-LDL, reduce intracellular lipid accumulation [31], and prevent the development of AS in apoE^−/−^ mice by reducing the aortic AS plaque area, which plays a positive role in preventing the development of AS [32]. Although circRNAs have been confirmed to play a role in AS pathogenesis, the exact effects of circRNAs in AS still remain unknown. Therefore, this study aimed to investigate the expression of AST-related genes during cholesterol efflux, and to explore the mechanism of circRNAs in this process, as well as the signaling pathways that interact with miRNAs and target genes.

## 2. Materials and Methods

### 2.1. Main Material

Astaxanthin standard (purity ≥ 97%) was purchased from Sigma-Aldrich (Saint Louis, MO, USA). The AST was completely dissolved in dimethyl sulfoxide (DMSO; Beijing Solarbio Science and Technology Co., Ltd., Beijing, China) to obtain an AST stock solution (50 mM), filtered using a 0.45 μm membrane, liquored, and stored at −20 °C. Human ox-LDL was purchased from Yiyuan biotechnologies (Guangzhou, China) [33,34]. Oil red O was obtained from Solarbio (Beijing, China).

### 2.2. Cell Cultures

RAW264.7 cells were obtained from the Cell bank of Type Culture Collection of Chinese Academy of Sciences (Shanghai, China) and were cultured in Dulbecco’s modified Eagle medium (DMEM) containing 10% fetal bovine serum (FBS), 100 units per mL penicillin, and 100 μg/mL streptomycin at 37 °C in a humidified atmosphere with 5% CO_2_. When the confluency reached 80%, Raw264.7 cells were treated with 50 μg/mL ox-LDL to form foam cells.

### 2.3. Oil Red O Staining

RAW264.7 cells were cultured in six-well sterile culture plates and treated with 50 μg/mL ox-LDL and concentration-escalating AST (0, 0.5, 5, and 50 μM) for 48 h. The procedure is outlined briefly as follows: after treatment, the medium was discarded, and the cells were washed three times with PBS and fixed with 4% PFA for 20–30 min. Then, the fixative was removed, and the cells were washed twice with distilled water and soaked in 60% isopropanol for 5 min. The isopropanol was discarded, freshly prepared ORO stain was added, and the cells were soaked for 10–20 min. Then, the cells were washed until there was no excess staining solution, distilled water was added to cover the cells, and red intracellular lipid droplets were observed under a microscope.

### 2.4. Measurement of Medium Total Cholesterol and Cell Total Cholesterol

After RAW264.7 cells were loaded with ox-LDL and different concentrations of AST (0, 0.5, 5, and 50 μM) for 24 h, 36 h, and 48 h, levels of medium total cholesterol and cell total cholesterol were detected according to the manufacturer’s instructions (Nanjing Jiancheng Bioengineering Institute, Nanjing, China).

### 2.5. Total RNA Extraction, cDNA Synthesis, and RT-PCR Analysis

Total RNA was extracted using Trizol reagent (Invitrogen, Gaithersburg, MD, USA). The quality and quantity of RNA were determined spectrophotometrically at optical density (OD)_260_ and OD_260_/OD_280_ = 1.8–2.1 using a NanoDrop spectrophotometer (Thermo Fisher Scientific, Waltham, MA, USA). For qRT-PCR analyses, 1 μg of total RNA was used with a reverse transcription kit (Roche, Shanghai, China) to obtain cDNA according to the manufacturer’s protocol. The specific primers of the mRNAs for mouse ABCA1, ABCG1, SR-BI, and glyceraldehyde-3-phosphate dehydrogenase (GAPDH) are listed in Table 1. PCR was performed using the following light cycle conditions: 95 °C for 10 min, followed by 40 cycles of 95 °C for 15 s, 56 °C for 30 s. The synthesis of polymerase chain reaction products was monitored by melting curve analysis and agarose gel electrophoresis. The quantitative results for mRNA and circRNA were normalized by glyceraldehyde-3-phosphate dehydrogenase and 18s rRNA, respectively. The relative quantification was calculated using the 2^−ΔΔCt^ method.

### 2.6. Western Blot Analysis

RAW264.7 cells were cultured on six-well plates with 10% FBS DMEM at a density of 1.0 × 10^6^ cells/well. When the RAW264.7 cells adhered, they were incubated with or without indicated treatment. After the cells were collected, they were washed twice with ice-cold phosphate-buffered saline (PBS). RIPA lysate and protease inhibitor 100:1 were used to extract the protein, and total protein concentration was calculated using a bicinchoninic acid (BCA) protein assay kit (Thermo Scientific, Waltham, MA, USA). The extracted proteins were separated on an 8% sodium dodecyl sulfate polyacrylamide gel electrophoresis (SDS-PAGE) gel at 100 V, and the protein was transferred from the gel to polyvinylidene difluoride (PVDF) membranes (Millipore, Billerica, MA, USA) at 250 mA for 150 min. The blots were then blocked for 1.5 h at room temperature with skimmed milk powder in 5% bovine serum albumin (BSA). The blots were incubated overnight shaker at 4 °C with mouse anti-ABCA1 (1:500, Abcam, Cambridge, MA, USA), rabbit anti-ABCG1 (1:500, Proteintech, Wuhan, China), rabbit anti-SR-BI (1:1000, Sangon Biotech, Shanghai, China), and rabbit anti-GAPDH (1:1000, Sangon Biotech, Shanghai, China). After rinsing three times for 30 min with TBST, the blots were incubated with horseradish peroxidase (HRP)-labeled goat anti-rabbit IgG (1:1000, Beyotime, Shanghai, China) or HRP-conjugated anti-mouse IgG (1:1000, CST, USA) for 70 min at room temperature. Lastly, the protein expression was detected using a chemiluminescence Western blotting system.

### 2.7. RNA Sample Preparation for High-Throughput Sequencing

At least 1.0 × 10^7^ cells were harvested from 50 μM AST treated and untreated groups. Total RNA was isolated with Trizol and reagent (Invitrogen, USA), and the linear RNAs were digested and removed with Rnase R. After removing the ribosomal RNA, the cDNA libraries of the circRNAs from each sample were generated according to Illumina standard protocols (Genergy Biotechnology, Shanghai, China). Then, transcriptome sequencing of RNA was conducted by the Illumina HiSeq 2500 platform according to the manufacturer’s instructions. Quality control was performed on primary sequencing data (raw reads), removing reads containing adapters, reads containing Ploy-N, and low-quality reads from raw data to filter out low-quality readings. Gene expression levels were normalized using the reads per kilobase transcriptome per million mapped reads method. CircRNAs with ∣log_2_ (fold change)| ≥2 and a *p*-value <0.05 were considered to be significantly differentially expressed. The circRNAs currently known were annotated with their circBase ID, while the new circRNAs were annotated with their gene symbol.

### 2.8. Functional Enrichment Analysis

To better comprehend the potential mechanisms of RCT, the Database for Annotation, Visualization, and Integrated Discovery (DAVID) (http://david.abcc.ncifcrf.gov/ (accessed on 15 August 2020)) was used to identify the most functional annotations and to predict the potential functions of all circRNAs. Gene Ontology (GO) enrichment analysis, including biological processes (BP), molecular functions (MF), and cellular components (CC), was used to provide gene extensive annotation and gene products (http://www.geneontology.org (accessed on 15 October 2020)), while Kyoto Encyclopedia of Genes and Genomes (KEGG) analyses were performed to understand high-level functions and interactions among differentially expressed genes in the pathways (http://www.genome.ad.jp/kegg/ (accessed on 20 October 2020)). The statistical significance threshold of GO term enrichment or the significance of the KEGG pathway correlation was set at *p* < 0.05.

### 2.9. Differential Expression Analysis of circRNAs

The back spliced reads per (BSRP) million mapped reads was used to normalize the differently expressed circRNAs in each sample. The *DESeq2* package in Bioconductor was used to analyze and compare the sequencing data of the treatment group and the control group. Using log_2_ (fold change) as a criterion, the top 10 circRNAs with the most significant upregulation and downregulation expression were selected and sequenced, as shown in Table 2.

### 2.10. Target miRNA of circRNAs Prediction

CircRNAs can bind miRNAs in various forms, mediate miRNA transcription, and play a key role in gene regulation. MiRNAs that were targeted by the circRNAs in the network were predicted using the miRanda software. We further combined the analysis of differentially expressed circRNAs with the target prediction of miRNAs, obtaining the top 92 miRNAs according to *p*-value sequencing, including 45 miRNAs associated with upregulated circRNAs and 47 miRNAs associated with downregulated circRNAs. The establishment of the circRNA–miRNA network allowed us to better explore the potential relationship between circRNAs and miRNAs in AST regulation of cholesterol efflux.

### 2.11. circTPP2 Knockdown and miR-3073b-5p Mimic/Inhibitor Transfection

The design and synthesis of small interfering RNAs (siRNAs), miR-3073b-5p mimic/inhibitor, and their negative controls were completed using GeenPharma (Shanghai, China). siRNAs, miR-3073b-5p mimic/inhibitor, and their negative controls were transfected into RAW264.7 cells using Lipofectamine 2000 (Invitrogen, Gaithersburg, MD, USA) after the cells were inoculated 1 day before transfection with a density of 30–50%. Then, 4–6 h after transfection, the medium was replaced with fresh medium. On the second day, 50 μg/mL ox-LDL and 50 μM AST were added, and the cells were collected to detect the silence effect after 48 h treatment.

### 2.12. Dual Luciferase Reporter Assay

To demonstrate the binding relationship among circRNA, miRNA, and mRNA, a luciferase assay was performed according to the protocol of the manufacturer. The sequences of circTPP2 or the 3′-UTR of the ABCA1 mRNA comprising the miR-3073b-5p-binding site were synthesized and inserted into PmirGLO vector (Promega, Madison, WI, USA) to construct the luciferase vector. Next, miRNA mimics/miR-NC and the above luciferase vector were co-transfected into 293T cells. The 293T cells were harvested at 48 h after transfection, and the relative luciferase activities were measured using the Dual Luciferase Reporter Assay Kit (Promega, WI, USA).

### 2.13. Statistical Analysis

Continuous variables are expressed as the mean ± standard error of mean (SEM). All experiments were repeated in triplicate. Statistical analyses were performed using GraphPad Prism version 5.0 (GraphPad Software, San Diego, CA, USA). Significant differences between the two groups were found using the Student *t*-test, and the means of more than two groups were compared using one-way ANOVA with Tukey’s post hoc test. A *p*-value <0.05 was considered statistically significant.

## 3. Results

### 3.1. Effect of AST on ox-LDL-Induced Foam Cell Formation and Promoting Cholesterol Efflux in RAW264.7 Cells

In order to evaluate the role of AST in the development of AS, Oil red O and total cholesterol measurement were used to detect the effects of AST on cholesterol efflux in RAW264.7-derived foam cells. Compared with the blank group, the number of red lipid droplets engulfed by RAW264.7 macrophages after 50 μg/mL ox-LDL-induced was significantly increased, indicating that the foam cell model was successfully established. At the same time, AST treatment markedly reduced the Oil Red O staining area of the cells in a concentration-dependent manner (Figure 1A). Then, the level of medium or intracellular TC was determined at 24 h, 36 h, and 48 h after treatment (Figure 1B,C). Compared with the control group, the levels of medium TC and cell TC in AST-treated cells gradually increased and decreased, respectively. AST treatment led to a dramatic reduction in cytoplasmic TC, while triggering medium TC accumulation. Parenthetically, we investigated whether ABCA1, ABCG1, and SR-BI were involved in the effects of AST on ox-LDL-induced RAW264.7 cells. Treatment with AST (0, 0.5, and 5 μM) significantly increased the mRNA level of ABCA1, ABCG1, and SR-BI compared with model (Figure 1D). Simultaneously, Western blot experiments were used to detect the expression of cholesterol efflux-related proteins, and the expression of ABCA1, ABCG1, and SR-BI protein remarkably increased in an AST concentration-dependent manner (Figure 1E–G). These results suggest that AST may stimulate cholesterol efflux by upregulating the expression of ABCA1, ABCG1, and SR-BI.

### 3.2. Differential Expression of circRNAs

To investigate the possible role of circRNAs in cholesterol efflux with AST, we estimated the expression levels of genome-wide-scale circRNAs in RAW264.7 macrophage-derived foam cells with or without AST treatment using microarray analysis. The volcano plot (Appendix A) and scatter plot (Appendix A) further display that the differential expression of circRNAs in AST treatment was significant (|log_2_ (fold change)| > 2). A total of 1751 circRNAs were identified as remarkably differentially expressed between the treatment cells and control cells, of which 826 circRNAs were upregulated and 925 circRNAs were downregulated (Appendix A).

### 3.3. Functional Analysis of Differentially Expressed circRNAs

To better comprehend the potential function of circRNAs, the linear transcripts of the corresponding origination genes for circRNAs were annotated, and Gene Ontology (GO) and Kyoto Encyclopedia of Genes and Genomes (KEGG) enrichment analyses were carried out to evaluate the major biological functions of differentially expressed circRNA–miRNA–mRNA network. The top 10 highly enriched GO terms of biological processes (BPs), cellular components (CCs), and molecular functions (MFs) are shown in Appendix A. This result revealed that, in the BP domain, the most meaningful enriched GO terms were phytosphingosine metabolic process, N-glycan processing to lysosome, and phytosphingosine biosynthetic process. For the CC domain, the most meaningful enriched GO terms were organelle, membrane-bounded organelle, and recycling endosome. In addition, among MFs, the most meaningful enriched GO terms were phosphotransferase activity, cystine–glutamate antiporter activity, and cystine secondary active transmembrane transporter activity. The GO dendrogram displays the top 10 significantly enriched in each subcategory (Appendix A). Moreover, only one signaling KEGG pathway was significantly enriched in the ceRNA network, namely, phosphonate and phosphinate metabolism (Appendix A). The results indicate that, in the protection of AS via the AST mechanism, the role of circRNA is closely related to the mechanistic target of metabolism signaling pathway, and further validation is required to confirm these results.

### 3.4. Validation of Candidate circRNAs by qRT-PCR

According to the fold-change of the deep sequencing results, the five top upregulated circRNAs (circZfp710, circVps13a, circTPP2, circAbraxas1, and circSnx9) and five top downregulated circRNAs (circRab3gap2, circRprd1b, circIde, circUbe2cbp, and circSclt1) were selected for validation using qRT-PCR. As shown in Figure 2, it was found that the expression of circZfp710 and circTPP2 was significantly upregulated, while circSclt1 and circRprd1b were downregulated compared with the control group, consistent with the sequencing results. However, three upregulated circRNAs (circVps13a, circAbraxas1, and circSnx9) and three downregulated circRNAs (circRab3gap2, circIde, and circUbe2cbp) were not significantly expressed or inconsistently expressed compared with the sequencing results. The above results indicate that these four significantly expressed circRNAs (circZfp710, circTPP2, circSclt1, and circRprd1b) may be involved in the process of AST promoting cholesterol efflux from macrophages.

### 3.5. Analysis of Differentially Expressed circRNAs Targeted miRNAs

By acting as an effective molecular sponge for miRNAs, cytoplasmic circRNA increased target gene expression; thus, we searched for miRNAs that might be adsorbed by these circRNAs using miRanda software. In this network, each miRNA corresponds to a differentially expressed circRNA. It was found that there was more than one miRNA targeted by differentially expressed circRNAs; hence, the top 45 miRNA targeting upregulated circRNAs and the top 47 miRNA targeting downregulated circRNAs according to *p*-value sequencing information were selected. The circRNA-targeting miRNA networks are shown in Figure 3. The interaction networks indicate the co-expression patterns of miRNAs, circRNAs, and mRNAs in ox-LDL-induced RAW264.7 macrophages.

### 3.6. siRNA Interference Screened Out the Most Differentially Expressed circRNAs

Two siRNAs for upregulated circRNAs (circZfp710 and circTPP2) were designed and transfected into RAW264.7 cells to knock down the expression of the target circRNA, and four groups (blank control, NC, siRNA 50 nM, and siRNA 100 nM) were set to screen the siRNA efficiency. After three repeated verifications, it was found that the interference effect was better when the siRNA concentration of the two circRNAs was 50 nM (Figure 4A). Therefore, 50 nM siRNA was used for subsequent experiments, continuing to interfere with these two circRNAs to verify the effects on the mRNA levels of ABCA1, ABCG1, and SR-BI. After circZfp710 was interfered with by siRNA, the expression levels of ABCA1 and SR-BI were increased, while the expression level of ABCG1 was decreased. The expression levels of ABCA1, ABCG1, and SR-BI did not change significantly after AST intervention. Results showed that ABCA1 levels decreased upon knockdown of circTPP2 and significantly increased in the AST intervention cell model, whereas changes in ABCG1 and SR-BI were not obvious (Figure 4B). On the basis of these observations, we focused on circTPP2, which demonstrated an association with the ABCA1 gene.

### 3.7. Identification and Characterization of circTPP2

In terms of the annotation in circBase (http://www.circbase.org/ (accessed on 15 September 2021)), circTPP2 was found to be generated from the exons of TPP2, which is located on chr1: 43953257–43956625 (Figure 5A). Meanwhile, Sanger sequencing of PCR amplicons was performed to further confirm the head-to-tail junction, which was consistent with circTPP2 annotation (Figure 5B). Then, we designed divergent and convergent primers to amplify the back-spliced (circTPP2) and linear products (TPP2), respectively. It was found that divergent primers could be detected in cDNA but not gDNA, indicating that circTPP2 was looped by back-splicing (Figure 5C).

### 3.8. Validation of Predicted miRNAs Related to circTPP2

Considering that circRNAs are capable of binding to certain miRNAs and, therefore, regulating downstream genes, we found miRNAs with binding sites to circTPP2 through the prediction of TargetScan, starBase, miRanda, and miRDB software. In the prediction, the top four miRNAs for circTPP2 sponging were miR-186-5p, miR-3073a-5p, miR-3073b-5p, and miR-6537-5p (Figure 6A). Hence, we verified their expression by qRT-PCR after circTPP2 was knocked down by the specific siRNA. Then, it could be found that the expression of miR-3073b-5p was significantly upregulated, while the expression of the remaining four miRNAs was differentially downregulated (Figure 6B). Therefore, miR-3073b-5p was identified as a possible candidate.

### 3.9. CircTPP2 Indirectly Bound to miR-3073b-5p to Regulate ABCA1 Expression

The potential binding site between miR-3073b-5p and circTPP2 is illustrated in Figure 6C, showing that miR-3073b-5p might be a downstream target of circTPP2. To further confirm that circTPP2 specifically binds to miR-3073b-5p to regulate the expression of ABCA1, a dual luciferase reporter assay was performed. CircTPP2 and ABCA1 mutants were constructed by site-directed mutation and cloned into the PmirGLO plasmid. As shown in Figure 6D, the luciferase reporter results showed that the luciferase activities of the circTPP2 wildtype reporter exhibited no significant changes when co-transfected with the miR-3073b-5p mimic compared with the control reporter or mutated luciferase reporter. However, the above qPCR results confirmed that circTPP2 knockdown could increase the expression of miR-3073b-5p.

At the same time, bioinformatics analysis showed that miR-3073b-5p had two binding sites matching ABCA1(Figure 7A,B). The results of the dual luciferase assay showed that miR-3073b-5p significantly reduced wildtype ABCA1 (ABCA1-WT) luciferase activity, whereas this effect was abrogated when the miR-3073b-5p-binding site with ABCA1-mut1 was mutated. In contrast, miR-3073b-5p had effects on both ABCA1-WT and ABCA1-mut2 (Figure 7C,D), indicating that miR-3073b-5p may play a role in AST’s promotion of cholesterol efflux by directly targeting ABCA1.

## 4. Discussion

Atherosclerotic heart disease, which has high morbidity and mortality, is the primary disease threatening public health in China. The initiation and progression of AS is driven by cellular oxidative stress, lipid metabolism disorder, inflammatory cell infiltration, vascular smooth muscle cell proliferation, endothelial dysfunction, and platelet activation [35,36]. The initial event is endothelial injury and accumulation of LDL-C in the subcutaneous space caused by endothelial injury [37]. Therefore, preventing the formation of foam cells is the key to avoiding AS. Recently, AST, which has strong antioxidant and anti-inflammatory ability, has attracted wide attention as a dietary bioactive factor for the prevention of AS [14,22]. In this study, we investigated the effect of AST on cholesterol efflux in macrophages RAW264.7 and its underlying mechanisms. Firstly, RAW264.7 cells were exposed to ox-LDL to establish a foam cell model. The results of the oil red O staining showed that a large number of red lipid dots were found in the cells and the cell volume increased, indicating that the model was successfully established. Subsequent experimental results showed that AST suppressed RAW264.7 macrophages trans-differentiation and retarded the accumulation of intracellular lipid in a concentration-dependent manner. In terms of the mechanism, AST promoted RCT by increasing the mRNA and protein expression levels of ABCA1, ABCG1, and SR-BI [38,39]. Furthermore, we demonstrated that the circTPP2/miR-3073b-5p/ABCA1 signaling pathway is involved in AST-induced ABCA1 expression and atheroprotective effects.

AST is known as a non-vitamin A carotenoid isolated from salmon and crustacean aquaculture, and it is believed to have powerful antioxidant and anti-inflammatory properties [22]. Due to its high free-radical-scavenging activity, AST’s antioxidant activity is 10 times higher than that of other carotenoids such as lutein, canthaxanthin, and β carotene, and 100 times higher than that of α-tocopherol [40]. Therefore, AST is called “super vitamin E”. Previous studies have reported that AST can reduce the levels of total cholesterol and LDL-C, increase the concentration of HDL cholesterol, and inhibit the expression of ox-LDL and apoA-I, which effectively intervenes with AS [23,41]. In the foam cell model induced by ox-LDL, AST could effectively upregulate the expression of ABCA1, ABCG1, and SR-BI, promote cholesterol efflux, reduce the formation of foam cells, and further prevent AS. A previous study from our group found that AST could reduce the area of aortic atherosclerotic plaque in ApoE^−/−^ mice, indicating that AST is beneficial to enhance the RCT process and prevent the development of AS. In clinical studies, AST can also inhibit LDL oxidation, increase HDL-C and adiponectin levels, promote RCT efficiency, and improve serum cholesterol profiles [14,21,23]. These findings confirm the role of AST in cholesterol efflux and are consistent with our previous observations.

CircRNA is a special ncRNA, which is an indispensable member in many biological processes [42,43,44]. CircRNA exists widely in human cells, and its expression sometimes even exceeds 10 times that of its linear transcription products [45]. It is formed by reverse shearing of pre-mRNA, which is different from the classical shearing mode of general linear RNA; therefore, it has the characteristics of stability, specificity, and conservation [44,46]. With the rapid development of experimental technology and bioinformatics analysis, more and more circRNAs have been discovered and identified, but high-throughput sequencing technology is still limited to the detection of circRNAs in mice, rats, and humans. However, the high-throughput sequencing technology combined with bioinformatics analysis has opened a new door for research into circRNAs. In this study, a cell model of AST promoting cholesterol efflux was constructed, and high-throughput sequencing revealed nearly 10,000 circRNAs with differential expression. According to the screening conditions (|log_2_ (fold change)| > 2), a total of 1751 circRNAs were obtained, of which 925 were upregulated and 826 were downregulated. Chromosome categorization revealed that the differentially expressed circRNAs were mainly generated from exons in various chromosomes, accounting for 96.7%. In order to verify these RNA-seq results, qRT-PCR was performed. The most significantly upregulated and downregulated circRNAs were selected for screening and verification, and four circRNAs were finally obtained consistent with the sequencing results. To better understand the potential function of the differentially expressed circRNAs, GO function analysis and KEGG enrichment analysis were performed on these circRNAs. The results showed that circRNAs were mainly related to phosphate and hypophosphate metabolism biological processes, providing a novel idea for further exploration.

TPP2 is a huge exopeptidase, which exists in the cytosol of most eukaryotic cells, and its main function is to degrade protein together with proteasome [47]. Overexpression of the TPP2 gene leads to accelerated cell growth, genetic instability, and resistance to apoptosis, while inhibition of TPP2 gene expression renders cells sensitive to apoptosis [48]. circTPP2 is a novel endogenous ncRNA that regulate genes at the post-transcriptional level. NCBI BLAST revealed that circTPP2 is composed of exons of gene TPP2, which are located on chromosome chr1: 43953257–43956625.

Protein-coding genes and circRNAs can regulate each other by competitively binding to miRNAs. These circRNAs sponge with miRNAs to release their target mRNAs, which are defined as ceRNAs. miRanda software was used to predict miRNAs that might be sponged by these circRNAs. In this circRNA–miRNA network, the top 45 miRNAs targeted by upregulated circRNAs and the top 47 miRNAs targeted by downregulated circRNAs were displayed according to the sequencing information of *p*-values. It was found that there is more than one miRNA targeted by differentially expressed circRNAs, highlighting that circRNAs can indirectly regulate the expression of downstream genes by sponging with different miRNAs.

In the present study, the expression of circTPP2 was further increased after AST treatment in ox-LDL-induced RAW264.7 macrophages, but the role of circTPP2 in AS was not reported. ABCA1, identified as a cholesterol transporter, plays a key role in mediating cholesterol efflux from macrophages to apoA-I [49,50]. Cholesterol efflux efficiency was significantly reduced by 30% in primary macrophages lacking ABCA1 [50,51]. Our results showed that AST could regulate ABCA1 by upregulating the expression level of circTPP2, thereby promoting cholesterol efflux and reducing intracellular lipid accumulation. CircRNAs acting as miRNA “sponges” represent the most common way of influencing downstream gene expression. The targeted miRNAs of circTPP2 were predicted and screened by bioinformatics technology, according to which miR-186-5p was reported to decrease the expression of insulin-like growth factor and induce neuron apoptosis in human neuroblasts [52]. Through bioinformatics prediction and luciferase reporter analysis, it was found that co-transfection with ABCA1-WT and miR-3073b-5p mimic could significantly reduce luciferase activities, which indicated that they could be targeted directly. However, luciferase reporter activities found that circTPP2 had no direct relationship with miR-3073b-5p. Further investigations are warranted to elucidate the molecular mechanism underlying circTPP2’s negative regulation of miR-3073b-5p expression. Therefore, our results showed that AST treatment induced a decrease in miR-3073b-5p expression and an increase in circTPP2 and ABCA1 expression, thus promoting cholesterol efflux and enhancing the ability of RCT in AS.

Despite new findings related to the antiatherosclerosis properties of AST and the circTPP2/miR-3073b-5p/ABCA1 axis in this study, there were still some limitations to this research. One of the limitations was that we did not include healthy patients or compare the expression between healthy controls and patients with AS. On the other hand, a number of cell types are involved in AS, including vascular smooth muscle cells, CD4^+^ T cells, and endothelial cells. These different cell types contribute to atherosclerotic plaque development, progression, and inflammation. Hence, another limitation of our experiment is that we only used a single cell type for verification. At the same time, we detected the expression of circRNAs in RAW264.7 cells treated with AST in vitro rather than in murine primary cells, which could have resulted in a difference in circRNA expression between cell lines and primary cells. Although RAW264.7 cells treated with ox-LDL were used in vitro to confirm the effect of AST in this research, the complex inflammatory microenvironment in atherosclerotic plaque should be further considered. We will fully consider these limitations and carry out in-depth research in the future. Altogether, our study can provide a theoretical basis concerning AST in the exploration and treatment of AS.

## 5. Conclusions

In summary, we identified a novel circTPP2 and determined its new role in regulating macrophage cholesterol efflux. Moreover, it was reported for the first time that circTPP2 exerted its function by downregulating miR-3073b-5p, thereby promoting the function of ABCA1. These findings may provide a novel angle from which to examine the effect of AST in preventing AS.

## Figures and Tables

**Figure 1 molecules-28-01701-f001:**
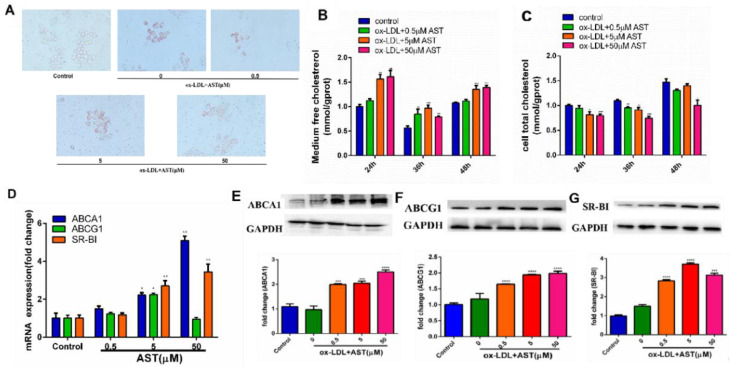
The effect of AST on ox-LDL-induced foam cell formation and cholesterol efflux in RAW264.7 cells. (**A**) RAW264.7 cells were incubated with AST (0.5, 5, and 50 μM) for 48 h, prior to being stimulated with ox-LDL (50 μg/mL). The formation of foam cells in each group was determined using Oil Red O staining. (**B**) The medium total cholesterol and (**C**) cell total cholesterol were detected using their corresponding commercial kits. qRT-PCR (**D**) and Western blot (**E**–**G**) experiments to detect the effects of different concentrations of AST on the expression of ABCA1, ABCG1, and SR-BI genes and protein levels after intervention. * *p* < 0.05 vs. control. ** *p* < 0.01 vs. control. *** *p* < 0.001 vs. control. **** *p* < 0.0001 vs. control.

**Figure 2 molecules-28-01701-f002:**
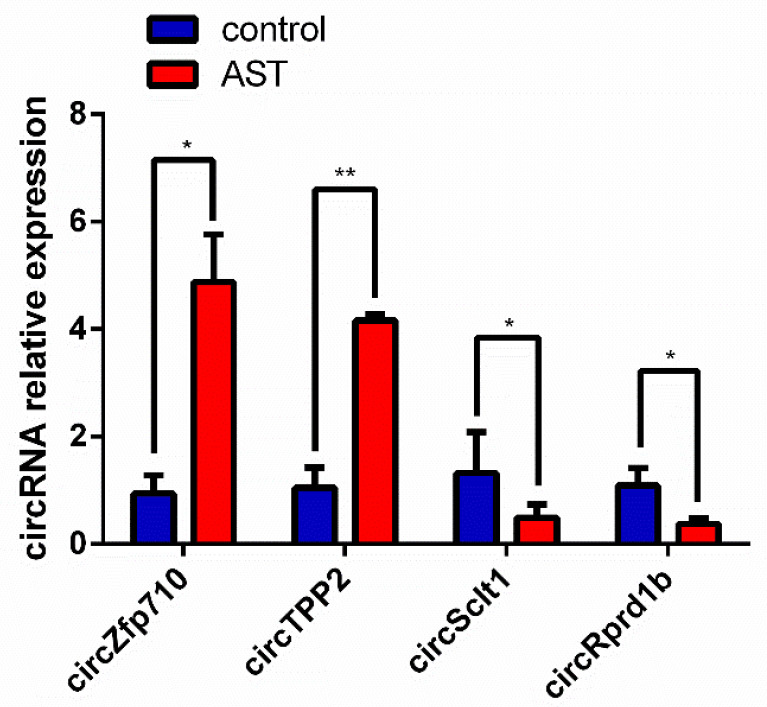
The four circRNAs that were differentially expressed in the process of AST promoting cholesterol efflux from macrophages. The expression of the first two circRNAs was significantly upregulated, and the expression of the next two circRNAs was significantly downregulated. These trends were consistent with RNA-seq results. Data are expressed as the mean ± SD and were analyzed by Student *t*-test. * *p* < 0.05 vs. control; ** *p* < 0.01 vs. control.

**Figure 3 molecules-28-01701-f003:**
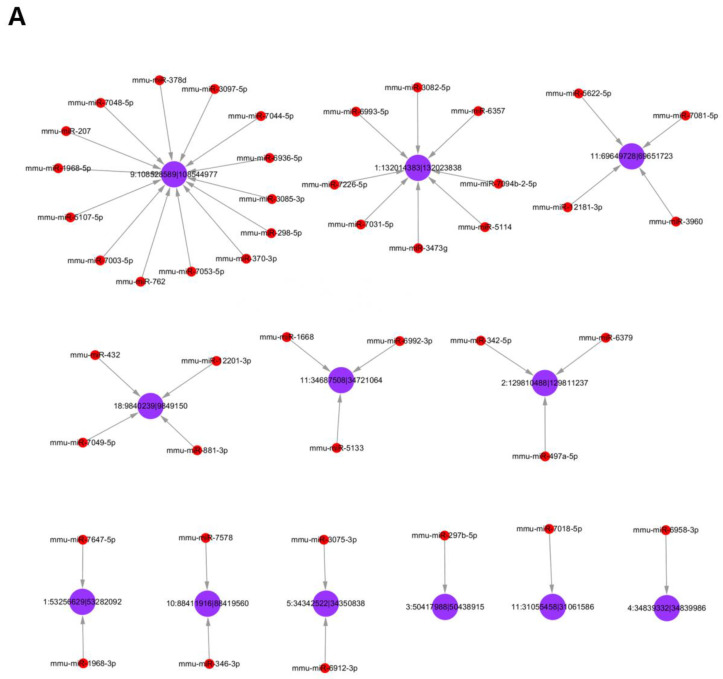
The circRNA-targeting miRNA networks. The top 45 putative target miRNAs of the upregulated circRNAs (**A**) and top 47 putative target miRNAs of the downregulated circRNAs (**B**) based on the *p*-value sorting information. The purple nodes represent the circRNAs, the red nodes represent the targeted miRNAs, and each gray line denotes a possible interaction of one circRNA with its targeted miRNA.

**Figure 4 molecules-28-01701-f004:**
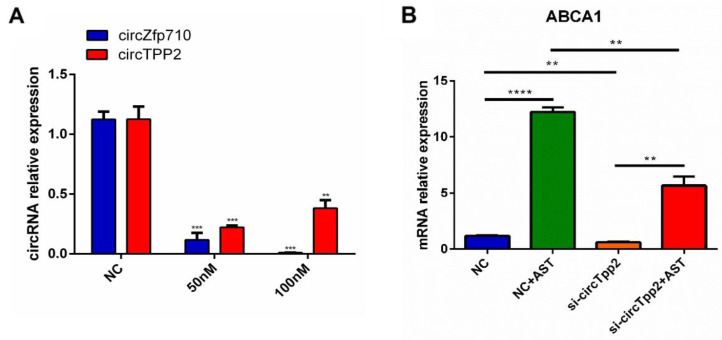
Effect of siRNA interference on the expression of circRNAs on target genes and proteins. (**A**) PCR to verify the interference effect of different concentrations of siRNA. (**B**) qRT-PCR to verify the effect of siRNA interfering with circTPP2 on ABCA1. ** *p* < 0.01 vs. control; *** *p* < 0.001 vs. control; **** *p* < 0.0001 vs. control.

**Figure 5 molecules-28-01701-f005:**
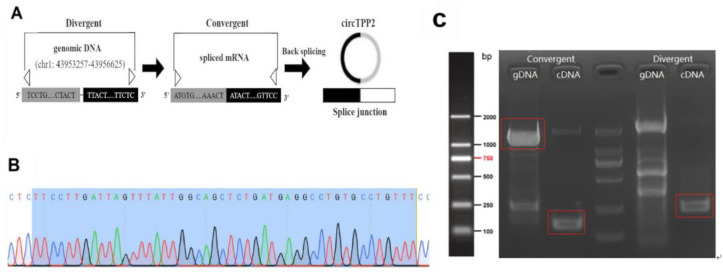
Identification of circTPP2 in RAW264.7 cells. (**A**) Genomic scheme of circTPP2. (**B**) The head-to-tail splicing junction associated with circTPP2 generation as identified by Sanger sequencing. (**C**) Agarose gel electrophoresis analysis of circTPP2 in cDNA and genomic DNA (gDNA) derived from RAW264.7 cells.

**Figure 6 molecules-28-01701-f006:**
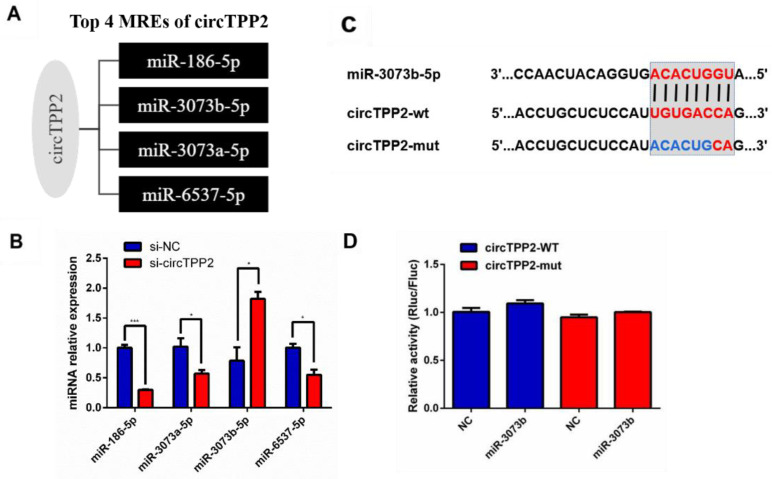
Validation of miRNAs for circTPP2 sponging. (**A**) Predicted top four miRNAs for circTPP2 by ceRNA analysis. (**B**) qRT-PCR to verify the expression of miRNAs after siRNA interfered with circTPP2. ** p* < 0.05, compared with the NC group; *** *p* < 0.001, compared with the NC group (**C**) Structure diagram of luciferase reporter of circTPP2 and its mutants. Seed match regions of circTPP2 and miR-3073b-5p are indicated as vertical lines. The mutation sites in circTPP2 are indicated in blue. (**D**) Luciferase assay for circTPP2. miR-3073b-5p mimics were co-transfected with wildtype circTPP2 (circTPP2-WT) or its mutations (circTPP2-mut) into 293T cells.

**Figure 7 molecules-28-01701-f007:**
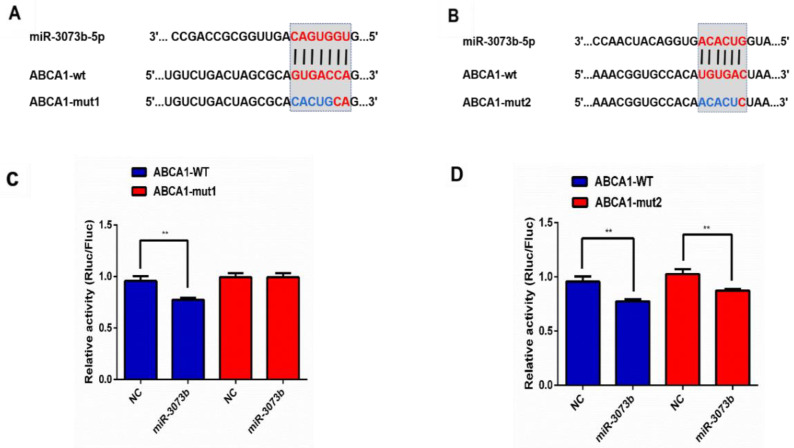
ABCA1 is a target gene of miR-3073b-5p. (**A**,**B**) Bioinformatics analysis indicating the two putative binding sites and corresponding mutant region for miR-3073b-5p within ABCA1. Seed match regions of ABCA1 and miR-3073b-5p are indicated as vertical lines. The mutation sites in ABCA1 are indicated in blue. (**C**,**D**) The dual luciferase reporter gene was used to verify the targeted relationship between miR-3073b-5p and ABCA1. The miR-3073b-5p mimics were co-transfected with ABCA1-WT or its mutations (ABCA1-mut1/ABCA1-mut2) into 293T cells. Luciferase activities were measured after 1 day of culture (*n* = 3 individual experiments). ** *p* < 0.01 vs. control.

**Table 1 molecules-28-01701-t001:** List of mainly primer sequences of the qRT-PCR.

Name	Primer Sequences (5′→3′)
GAPDH	F: AAGAAGGTGGTGAAGCAGG
	R: GAAGGTGGAAGAGTGGGAGT
ABCA1	F: CCAGAAGGGAGTGTCAGAAAT
	R: GGGAAACAGCCCAGTCAGTA
ABCG1	F: GAACTGCCCTACCTACCACAAC
	R: AAAGAAACGGGTTCACATCG
SR-BI	F: ACCCTAACCCAAAGGAGCAT
	R: CCACAGCAACGGCAGAACTA
U6	F: CTCGCTTCGGCAGCACATATACT
	R: ACGCTTCACGAATTTGCGTGTC
circZfp710	F: GGACAGCCAAGAAAUGCCCTT
	R: GGGCAUUUCUUGGCUGUCCTT
circTPP2	F: CAGAGCUGCCAAUAAACUATT
	R: UAGUUUAUUGGCAGCUCUGTT
circSclt1	F: GAAAACAUCUUAGAACAGCTT
	R: GCUGUUCUAAGAUGUUUUCTT
miR-6537-5p	F: CGCGGTGAGTTTCTCCCACT
	R: AGTGCAGGGTCCGAGGTATT
18s RNA	F: GTAACCCGTTGAACCCCATT
	R: CCATCCAATCGGTAGTAGCG
miR-186-5p	F: CGGGCCAAAGAATTCTCCTT
	R: CAGCCACAAAAGAGCACAAT
miR-3073a-5p	F: CGGGCGTGGTCACAGTTGGC
	R: CAGCCACAAAAGAGCACAAT
miR-3073b-5p	F: CGGGCATGGTCACAGTGGAC
	R: CAGCCACAAAAGAGCACAAT

**Table 2 molecules-28-01701-t002:** Specific information (circRNA ID, circBase ID, log_2_FC, and gene symbol) of the top five upregulated and top five downregulated circular RNAs.

circRNA ID	circBase ID	log2FC	Gene
7:80081049|80086608	—	4.392317423	Zfp710
19:16710271|16725693	—	4.087462841	Vps13a
1:43953257|43956625	—	3.906890596	TPP2
5:100805898|100812202	mm10_circ_001838	3.906890596	Abraxas1
17:5886997|5899495	—	3.906890596	Snx9
1:185266901|185272642	—	−4.857980995	Rab3gap2
2:158035996|158058827	mm10_circ_016411	−4.754887502	Rprd1b
19:37325148|37330612	—	−4.523561956	Ide
9:86422841|86448853	—	−4.392317423	Ube2cbp
3:41703386|41730919	—	−4.087462841	Sclt1

## Data Availability

Data are available on request from the authors.

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
