# Peer review of "Astaxanthin Alleviates Foam Cell Formation and Promotes Cholesterol Efflux in Ox-LDL-Induced RAW264.7 Cells via CircTPP2/miR-3073b-5p/ABCA1 Pathway"

_molecules, 2023, doi:10.3390/molecules28041701_

Round 1
Reviewer 1 Report
1.Line 127 “After the RAW264.7 cel” should be deleted.
2.Please explain“ why the level of medium TC at 36 h after treatment is lower than that at 24 h after treatment”?
3.Line228-229 “ Treatment with AST (0, 0.5, 5, 50 μM) significantly creased the mRNA level of ABCA1, ABCG1 and SR-BI compared with model (Figure 1D)”. But Figure 1D didn’t show Treatment with AST (50 μM) significantly creased the mRNA level of ABCG1 compared with model, please explain why.
4.Double check the manuscript and avoid the description in first person as "we".
5.Double check the manuscript and avoid too much similarity between introduction and discussion.
Author Response
Thanks for your comments concerning our manuscript molecules-2151618. Those comments are all valuable and very helpful for revising and improving our paper, as well as the important guiding significance to our researches. The revised portion is marked in red on the paper. Point by point responses to the reviews’ comments are listed in attachment.

Reviewer 2 Report
This article is devoted to the study of the molecular mechanisms of action astaxanthin. The work makes a good impression with its well thought-out design and careful execution of molecular biological studies. Nevertheless, the article has a number of shortcomings, which need to be corrected when finalizing it. That said, there are a number of issues that need to be clarified, as follows:
1. The paper does not characterize the ox-LDL used, which was purchased from Yiyuan biotechnologies (Guangzhou, China). Are there any literature data on the composition of these ox-LDLs, especially the oxidized products they contain? This is the weakest point of the article. It would probably be correct to obtain ox-LDL by the initiated free-radical oxidation method itself and characterize them by the level of lipoperoxides, MDA-modified LDL etc.
2. Undoubtedly, unoxidized LDL should have been used as a control. Why was this not done?
3. Why was the AST stock solution filtered by 0.45 μm membrane? It was not a true solution or suspension ?
4. Is there any work that has shown the antioxidant effect of astaxanthin in free-radical oxidation of LDL ? If so, why is this specific information not listed in the literature and not used in the discussion of the results ?
I believe that the answers to the questions raised should be included in the version of the article when it is finalized.
Author Response

(The authors gave the same response as above.)

Reviewer 3 Report
Title;Astaxanthin Alleviates Foam Cell Formation and Promotes Cholesterol Efflux in Ox-LDL-Induced RAW264.7 Cells via CircTPP2/miR-3073b-5p/ABCA1 Pathway
Comments; In my view, the results obtained in this study are worthy for publication. The manuscript needs major essential revision before publication. I would like to overview the revised version of the manuscript. I have the following comments/suggestions for authors to address before final decision on the manuscript.
1. Do there any PDB ID for circTPP2? What is the length of TPP2 exopeptide?
2. Objective of your study is not clear in the abstract.
3. Authors have suggested to give more information that how and by which mechanism the circTPP2 plays role in the regulation of macrophage cholesterol efflux.
4. Author has suggested to rewrite the table legend in Table 2.
5. Authors have been suggested to remove the spelling, spacing, and grammatical errors from the article.
6. Authors have to provide the future aspects of the analysis and their impact on the frontier of knowledge on the topic.
7. Figures should be provided in higher resolution and in a readable form.
8. Authors have to compare the findings with similar studies. Thereby lacks validation.
9. In the Introduction section the author should refer to the research paper and comment on recent in-silico techniques. It will be good information for the readers. I would like to recommend several papers, among many others, providing further explanation on this topic: PMID: 27194485 PMID: 28681927 PMID: 36307910 PMID: 35486518 PMID: 23782055 PMID: 31903852 PMID: 35276295 PMID: 32448055 PMID: 35604288
11. The authors have clearly mentioned the aim of the study in the last paragraph of the Introduction section. However, the objectives of the study are missing.
12. Lines 84 to 89 should be placed under an appropriate subheading.
13. Present the limitations of the study in the last paragraph of the Discussions section.
14. Elaborate on the summary section.
15. Figure 5A: Redraw the figure by not altering the original aspect ratio.
16. Conclusion section lacks proper data and should be elaborated based on results.Authors have to elaborate on the conclusion of their findings, by adding the major findings. The conclusion also lacks the future aspects of the study.
17. Information is incomplete in line 127.
18. “cular smooth muscle cell proliferation, endothelial dysfunction, platelet activation[33,34], and so on.” What authors meant by so on. They need to be specific.
19. “The circRNAs targeted miRNA network are shown in Figure 3.” Merely depiction is not enough. Authors provide insight about the probable findings in the text also.
20. “Moreover, we reported for the first time that circTPP2 exerted its function by regulating miR-3073b-5p,” Authors need to specifically mention how circTPP2 regulates mir-3073b-5b. Is it inhibiting its function or may increase its function?
Author Response

(The authors gave the same response as above.)
